# Enhanced Electrochemical Performance of PEO-Based Composite Polymer Electrolyte with Single-Ion Conducting Polymer Grafted SiO_2_ Nanoparticles

**DOI:** 10.3390/polym15020394

**Published:** 2023-01-11

**Authors:** Xuan Liu, Wanning Mao, Jie Gong, Haiyu Liu, Yanming Shao, Liyu Sun, Haihua Wang, Chao Wang

**Affiliations:** 1Key Laboratory of Auxiliary Chemistry and Technology for Chemical Industry, Ministry of Education, College of Chemistry and Chemical Engineering, Shaanxi University of Science and Technology, Xi’an 710021, China; 2Shaanxi Key Laboratory of Chemical Additives for Industry, Shaanxi University of Science and Technology, Xi’an 710021, China; 3Shaanxi Collaborative Innovation Center of Industrial Auxiliary Chemistry & Technology, The Youth Innovation Team of Shaanxi Universities, Shaanxi University of Science and Technology, Xi’an 710021, China

**Keywords:** composite solid electrolyte, single-ion conducting polymer grafting, SiO_2_ modification, RAFT polymerization, all-solid-state battery

## Abstract

In order to enhance the electrochemical performance and mechanical properties of poly(ethylene oxide) (PEO)-based solid polymer electrolytes, composite solid electrolytes (CSE) composed of single-ion conducting polymer-modified SiO_2_, PEO and lithium salt were prepared and used in lithium-ion batteries in this work. The pyridyl disulfide terminated polymer (py-ss-PLiSSPSI) is synthesized through RAFT polymerization, then grafted onto SiO_2_ via thiol-disulfide exchange reaction between SiO_2_-SH and py-ss-PLiSSPSI. The chemical structure, surface morphology and elemental distribution of the as-prepared polymer and the PLiSSPSI-*g*-SiO_2_ nanoparticles have been investigated. Moreover, CSEs containing 2, 6, and 10 wt% PLiSSPSI-*g*-SiO_2_ nanoparticles (PLi-*g*-SiCSEs) are fabricated and characterized. The compatibility of the PLiSSPSI-*g*-SiO_2_ nanoparticles and the PEO can be effectively improved owing to the excellent dispersibility of the functionalized nanoparticles in the polymer matrix, which promotes the comprehensive performances of PLi-*g*-SiCSEs. The PLi-*g*-SiCSE-6 exhibits the highest ionic conductivity (0.22 mS·cm^−1^) at 60 °C, a large t_Li+_ of 0.77, a wider electrochemical window of 5.6 V and a rather good lithium plating/stripping performance at 60 °C, as well as superior mechanical properties. Hence, the CSEs containing single-ion conducting polymer modified nanoparticles are promising candidates for all-solid-state lithium-ion batteries.

## 1. Introduction

Developing solid-state lithium-ion batteries with higher energy density and higher safety is one of the main research and development goals of next-generation batteries. Since Wright et al. [1] discovered that mixed polyethylene oxide (PEO) and alkali metal salts show ionic conductivity in 1973, solid polymer electrolytes (SPEs) have received extensive attention in lithium-ion batteries owing to their high flexibility, processability, shape versatility, and light weight [2]. Despite that various other polymers, such as poly(methyl methacrylate) (PMMA), poly(acrylonitrile) (PAN) and poly(vinylidene difluoride) (PVDF) are used as SPEs in lithium-ion batteries nowadays, PEO is still the focus of extensive research owing to its relatively low melting point (*T_m_*), low glass transition temperature (*T_g_*) [3,4], as well as its superior ability to dissolve large amounts of alkali metal salts [5]. However, linear PEO is typically in its semicrystalline state at room temperature, which may hamper the mobility of lithium ions [6]. To address this problem and to promote the compatibility of nanoparticles and PEO, composite solid electrolytes (CSEs) are fabricated by introducing polymer functionalized nanoparticles into PEO [7,8,9], which can help to decrease the crystallinity of polymer chains and to increase the dispersion of nanoparticles in PEO, and consequently enhance the ionic conductivity as well as the mechanical strength of electrolytes.

Similar to commercial liquid electrolytes, most polymer electrolytes are double-ion conductors, and the lithium-ion transference number is low (t_Li+_ < 0.5). The anion as the main carrier can cause severe concentration polarization and can increase the internal resistance of the battery. This can result in voltage drop and severe side reactions, which would seriously affect the performance of solid-state batteries [10]. The polymer electrolyte obtained by chemically linking the anion to the polymer backbone or introducing anion acceptors to limit the movement of the anions is called a single-ion conducting solid polymer electrolyte (SIC-SPEs). The t_Li+_ of SIC-SPEs is usually close to 1, and SIC-SPEs can effectively suppress the concentration gradient of anions, reduce the internal impedance and increase the discharge voltage [11,12,13]. According to the model proposed by Chazalviel [14] electrolytes with t_Li+_ close to 1 can weaken the driving force for dendrites formation. Therefore, SIC-SPEs can avoid the disadvantages of traditional double-ion conductors, such as the formation of lithium dendrites and poor battery cycle performance, and are regarded as promising candidates for the electrolyte of all-solid-state batteries.

Strategies to combine the advantages of both functionalized nanoparticles and SIC polymers have been proposed. PEO-based CSE containing nanoparticles of SiO_2_ and Al_2_O_3_ modified with lithium [(4-methylphenyl)sulfonyl][(trifluoromethyl) sulfonyl]amide or a mixture of this lithium salt with poly(ethylene glycol) strands exhibit a high ionic conductivity (0.19 mS·cm^−1^ at 70 °C) and a high electrochemical stability (up to 5 V vs. Li^+^/Li) [15]. Solid electrolytes mixing lithium single-ion polymer (lithium (4-styrenesulfonyl) (trifluoromethanesulfonyl) imide (LiSTFSI)) grafted onto the surface of Ta-doped garnet Li_6.4_La_3_Zr_1.4_Ta_0.6_O_12_ (Li@LLZTO) nanoparticles with PEO show a high lithium ion conductivity (1.50 mS·cm^−1^) at 60 °C, a wide electrochemical window (~5.3 V vs. Li^+^/Li) and a large t_Li+_ (0.77) [16]. Single-ion conducting polymeric nanoparticles (SIC-PNPs) are also used as functional fillers to promote the performances of PEO. SIC-PNPs prepared through emulsion polymerization of styrene and divinylbenzene with a precise size used in CSEs enable high ionic conductivities ranging from 0.2 to 1 mS·cm^−1^, high t_Li+_ of 0.99, high electrochemical stabilities up to 5.5 V (vs. Li^+^/Li), and stable storage moduli of >10 MPa at 25–90 °C [17]. Lithium sulfonamide surface-functionalized polymer nanoparticles prepared through a semibatch emulsion polymerization with the help of cross-linked poly(methyl methacrylate) deliver a significant stiffening effect on the PEO matrix (E′ > 10 MPa at 80 °C) while the CSEs retain high ionic conductivity values (σ = 0.66 mS·cm^−1^) [18].

In this study, a PEO-based composite solid electrolyte with PLiSSPSI-*g*-SiO_2_ (poly (lithium (4-styrenesulfonyl)(phenylsulfonyl) imide)-*grafted*-SiO_2_) nanoparticles (PLi-*g*-SiCSE) for solid-state lithium-ion battery was designed, with PEO as matrix and functionalized SiO_2_ as filler, in which PLiSSPSI was synthesized by reversible addition-fragmentation chain transfer (RAFT) polymerization, then simply grafted onto SiO_2_ via thiol-disulfide exchange reaction. PEO/LiTFSI CSE was also prepared as the control. In terms of the single-ion conducting polymer layer on the surface of nanoparticle, the charge distribution on the surface of SiO_2_ can be adjusted and the interfacial compatibility between SiO_2_ and PEO can be also promoted, which results in the increased content of amorphous regions in polymer matrix. Hence, the PLi-*g*-SiCSE exhibits higher ionic conductivity, larger t_Li+_, wider electrochemical stability window, and better cycling stability. This demonstrates that PLi-*g*-SiCSE has great potential for all-solid-state lithium batteries.

## 2. Materials and Methods

### 2.1. Materials

Sodium 4-vinylbenzenesulfonate (Macklin, ≥98%), thionyl chloride (SOCl_2_, Macklin, 99%), benzenesulfonamide (Macklin, 98%), triethylamine (Macklin, 99%), anhydrous LiOH (Macklin, 98%), 2-methyl-2-[(dodecylsulfanylthiocarbonyl)sulfanyl]propanoic acid (DTPA, Macklin, 98%), 2,2′-azobis(2-methylpropionitrile) (AIBN, Macklin, 98%), (3-mercaptopropyl)trimethoxysilane (MPTMS, Macklin, 97%), 2,2′-dipyridyldisulfide (Macklin, 98%), methanol (Macklin, 99.5%), ethanol (Macklin, 99.5%), acetonitrile (Macklin, 99.5%), dimethyl formamide (DMF, Macklin, 99.5%), dichloromethane (DCM, Macklin, 99.5%), ethyl acetate (Macklin, 99.8%), hexane (Macklin, 99.8%), poly(ethylene oxide) (PEO, *M_w_* = 600,000 g·mol^−1^, Macklin), 4-dimethylaminopyridine (DMAP, Aladdin, 99%), 2-mercaptoethanol (Aladdin, 98%), *N*,*N*’-dicyclohexylcarbodiimide (DCC, Aladdin, 99%), NH_4_OH solution (Aladdin), glacial acetic acid (Aladdin, 99.7%), anhydrous MgSO_4_ (Aladdin, AR), NaHCO_3_ (Aladdin, AR), hydrochloric acid (Aladdin, AR), bis(trifluoromethane)sulfonimide lithium (LiTFSI, Aldrich, 99.95%), ultrapure water was prepared using ultra-pure water equipment (Beijing Cheng Ruiyuan Technology Co., Ltd., Beijing, China).

### 2.2. Preparation of Single-Ion Conducting Polymer through RAFT Polymerization

#### 2.2.1. Synthesis of Lithium (4-styrenesulfonyl)(phenylsulfonyl) imide (LiSSPSI) Monomer [19]

A solution of thionyl chloride (SOCl_2_, 208.20 g, 1.75 mmol) in anhydrous dimethyl formamide (DMF, 85 mL) was stirred in a double-neck flask immersed in an ice bath for 30 min. Sodium 4-vinylbenzenesulfonate (51.55 g, 0.25 mol) was progressively added into the solution during 30 min and the mixture was stirred for 2 h at 0 °C, then for 12 h at room temperature. After pouring the mixture into ice water (200 mL) to remove the unreacted SOCl_2_, dichloromethane (DCM, 150 mL) was then added and the product was extracted in the organic phase after treating with ice water (3 × 100 mL). The organic phase was dried over anhydrous MgSO_4_, then concentrated with a rotary evaporator after filtration to yield a yellow bright liquid 4-styrenesulfonyl chloride, which was stored in dry acetonitrile under an argon gas atmosphere at 0 °C.

Benzenesulfonamide (28.22 g, 0.18 mol), 4-dimethylaminopyridine (5.0 g, 0.04 mol) and triethylamine (27.25 g, 0.27 mol) were dissolved in dry acetonitrile (200 mL) under an argon gas atmosphere and stirred at 0 °C for 30 min. Then, the solution of 4-styrenesulfonyl chloride in dried acetonitrile was added slowly into the mixture via syringe for 20 min. Afterwards, the mixture was kept at 0 °C for 1 h, then stirred for 24 h at room temperature. After removing the solvent with a rotary evaporator, the resulting brown viscous liquid was dissolved in DCM. The solution was then washed with 0.5 M aqueous solution of NaHCO_3_ and 1.0 M hydrochloric acid for three times. The acid monomer was obtained after removing DCM, then neutralized with LiOH to produce LiSSPSI monomer.

^1^H NMR (400 MHz, D_2_O): δ = 7.47 (Ph, 5H), 7.34-7.22 (Ph′, 5H), 6.70 (CH_2_=C*H*, 1H), 5.84 (CH*H*=CH, 1H), and 5.36 (C*H*H=CH, 1H).

#### 2.2.2. Synthesis of 2-Methyl-2-[(dodecylsulfanylthiocarbonyl)sulfanyl]propanoic acid-2-(2-pyridyldithio) ethyl ester (Py-ss-DTPA) [20]

Initially, 2-(2-Pyridinyldithio)ethanol (PDE) was prepared as follows [21]. The 2,2′-dipyridyldisulfide (9.90 g, 45.0 mmol) and glacial acetic acid (0.66 mL) were dissolved in methanol (90 mL) under N_2_ atmosphere. Then, a solution of 2-mercaptoethanol (1.65 g, 22.1 mmol) in methanol (10 mL) was added dropwise into the mixture. The solution was stirred for 5 h at room temperature. A light-yellow oil was obtained after removing solvent with rotary evaporator and was purified by column chromatography on silica gel with a mixture of ethyl acetate: hexane (1:2/*v:v*).

To a solution containing 2-methyl-2-[(dodecylsulfanylthiocarbonyl)sulfanyl]propanoic acid (DTPA, 0.9 g, 2.23 mmol), 2-(2-pyridinyldithio)ethanol (0.4 g, 2.14 mmol), 4-dimethylaminopyridine (26.1 mg, 0.214 mmol) and dry dichloromethane (20 mL), *N*,*N*’-dicyclohexylcarbodiimide (0.44 g, 2.14 mmol) were added dropwise, and the mixture was stirred overnight at room temperature. After filtration, the resulting product py-ss-DTPA was obtained, and further purified by column chromatography on silica gel with a mixture of ethyl acetate: hexane (1:8/*v:v*).

^1^H NMR (400 MHz, CDCl_3_): δ = 8.52 (Ph, 1H), 7.83-7.67 (Ph′, 2H), 7.15 (Ph″, 1H), 4.41 (C*H*_2_-O, 2H), 3.32 (S=C-S-C*H*_2_, 2H), 3.08 (S-S-C*H*_2_, 2H), 1.73 (C-(C*H*_3_)_2_, (CH_2_)_9_-C*H*_2_,8H), 1.30 (-(C*H*_2_)_9_-, 18H), 0.92 ((CH_2_)_9_-C*H*_3_, 3H).

#### 2.2.3. Synthesis of PLiSSPSI with Pyridyl Disulfide Terminal Group (py-ss-PLiSSPSI)

Pyridyl disulfide terminated PLiSSPSI was prepared through RAFT polymerization of LiSSPSI using Py-ss-DTPA as the chain transfer agent. A mixture of LiSSPSI (3.29 g, 10 mmol), AIBN (50 mg, 0.3 mmol) and Py-ss-DTPA (171 mg, 0.32 mmol) in DMF (7.8 g) was degassed via three freeze-pump-thaw cycles, backfilled with argon and placed in an oil bath at 70 °C. After 16 h, the reaction was stopped by cooling in liquid nitrogen. Conversion was determined by ^1^H NMR in D_2_O. The polymer was dried in vacuo. (*M*_n,theo_ = 10,000 g·mol^−1^, *M*_n,SEC_ = 10,100 g·mol^−1^, Ð = 1.7).

### 2.3. Preparation of PLiSSPSI Grafted SiO_2_ Nanoparticles (PLiSSPSI-g-SiO_2_)

#### 2.3.1. Synthesis of Thiol-Functionalized Silica Particles (SiO_2_-SH) [22,23]

MPTMS (5 mL) was dissolved in water (500 mL) with mechanical stirring, a certain amount of NH_4_OH solution was added into the mixture to adjust pH to 11. After that, the mixture was stirred at room temperature for 24 h. The resulting microspheres were purified by repeated centrifugation and redispersion cycles, replacing supernatants with ethanol and H_2_O, respectively. Finally, the SiO_2_-SH microspheres were redispersed in water by repeated ultrasonication for a few hours.

#### 2.3.2. Synthesis of PLiSSPSI-*g*-SiO_2_ Nanoparticles

PLiSSPSI-*g*-SiO_2_ nanoparticles were prepared by thiol-disulfide exchange reaction between thiols on SiO_2_-SH and py-ss-PLiSSPSI. py-ss-PLiSSPSI (1.0 g) and SiO_2_-SH (2.0 g) were dispersed in DMF (40 mL) under N_2_ atmosphere. The exchange reaction was performed at 60 °C for 12 h after degassed by three freeze-pump-thaw cycles. The PLiSSPSI-*g*-SiO_2_ nanoparticles were collected after centrifugation (10,000 rpm, 10 min), and washed with methanol. The PLiSSPSI-*g*-SiO_2_ nanoparticles were redispersed in ethanol.

### 2.4. Preparation of PEO-Based Composite Solid Electrolyte with PLiSSPSI-g-SiO_2_ Nanoparticles

A certain amount of PEO and LiTFSI ([EO]: [Li^+^] = 20) were mixed for stirring with a mass ratio of 2, 6 and 10 wt% PLiSSPSI-*g*-SiO_2_ nanoparticles (based on the mass of PEO), then the composite solid electrolytes were prepared by the solution casting method and dried in a vacuum oven at 60 °C for 48 h. After that, the films were pressed into a round piece with a diameter of 19 mm and stored in glove box. The corresponding composite solid electrolytes were denoted as PLi-*g*-SiCSE-x, where x represents the weight content of PLiSSPSI-*g*-SiO_2_ nanoparticles.

### 2.5. Characterization

The ^1^H NMR spectra were recorded on a nuclear magnetic resonance spectrometer (NMR, AMX-400, Bruker, Karlsruhe, Germany) with CDCl_3_ or D_2_O as the solvent. The molecular weight was measured by size exclusion chromatography (SEC, 1260 Infinity Ⅱ, Agilent, Santa Clara, CA, USA) with the PS standard for calibration and THF as the eluent. Fourier-transform infrared spectroscopy was recorded on a spectrophotometer (FT-IR, INVENIO, Bruker, Germany) in a range of 4000–400 cm^−1^ employing KBr pellets. UV-Vis analyses were performed on a Shimadzu UV-2600 spectrometer (UV-Vis, Shimadzu, Kyoto, Japan) using quartz cuvettes with 10 mm path length. The surface elemental and surface functional groups of SiO_2_-SH and PLiSSPSI-*g*-SiO_2_ were analyzed by X-ray photoelectron spectroscopy (XPS, AXIS SUPRA, Shimadzu, Japan). DLS was conducted on a Zetasizer Nano ZS90 (Malvern Instruments, Ltd., Malvern, UK) at a scattering angle of 90° (25 °C), SiO_2_-SH and PLiSSPSI-*g*-SiO_2_ were dispersed in ethanol under ultrasonication at 25 °C before measurement. The surface nanostructures of SiO_2_-SH and PLiSSPSI-*g*-SiO_2_ were recorded on a FEI Verios 460 SEM (FEI, Hillsboro, OR, USA) at an operating voltage of 2.0 kV. Elemental mapping was further performed using an analytical TEM instrument (FEI Tecnai G2 F20, FEI, USA) equipped with an energy-dispersive spectroscopy (EDS) system. The crystallization of PLi-*g*-SiCSE-x were analyzed by X-ray diffraction (XRD, D/max 2200PC, Tokyo, Japan) equipped with CuKα radiation, the angle of 2θ ranged from 10° to 80°, and the scan rate was 5° min^−1^. Glass transition temperature (*T_g_*) of Pli-*g*-SiCSE-x was recorded by differential scanning calorimetry (DSC, DSC7020, HITACHI, Japan) at a heating rate of 10 °C min^−1^ under nitrogen. The mechanical properties of Pli-*g*-SiCSE-x were studied by the stress-strain test using Hengyi HY-0580 universal testing machine with a 500 N loading cell at the tensile rate of 30 mm min^−1^.

SS|Pli-*g*-SiCSE-x|SS symmetric cells (SS = stainless steel) were assembled for ionic conductivity measurement. The ionic conductivities of Pli-*g*-SiCSE-x were measured by electrochemical impedance spectroscopy (EIS) by using electrochemical working station (PARSTAT 3000A DX, PAR, USA) with the frequency range from 1 MHz to 0.1 Hz and temperature between 25 and 80 °C. The ionic conductivity was calculated as follows:(1)$$\sigma =\frac{L}{R\cdot S}$$
in which σ is the ionic conductivity, R is the bulk resistance, L is the thickness of the Pli-*g*-SiCSE-x, and S is the contact area of SS electrode and Pli-*g*-SiCSE-x.

Li|Pli-*g*-SiCSE-x|Li symmetric cell was used to determine t_Li+_ of the Pli-*g*-SiCSE-x membrane based on EIS with a frequency range from 1 MHz to 0.1 Hz and with the alternating voltage amplitude of 10 mV at 60 °C. The t_Li_^+^ was measured as Equation (2):(2)$$t_{Li^{+}}=\frac{I_{S}\left(\Delta V-I_{0}R_{0}\right)}{I_{0}\left(\Delta V-I_{S}R_{S}\right)}$$
where ΔV, I_S_, and I_0_ are the polarization potential, steady-state current, and initial-state current, respectively. R_S_ and R_0_ represent the steady-state resistance and initial-state resistance.

The electrochemical stability of PLi-*g*-SiCSE-x was evaluated by linear sweep voltammetry (LSV) using a SS|PLi-*g*-SiCSE-x|Li cell with a scan rate of 0.1 mV s^−1^ from 0 to 6 V (vs. Li/Li^+^) at room temperature.

The charge/discharge performance of LFP|PLi-*g*-SiCSE-x|Li batteries (LFP = LiFePO_4_) was carried out on a battery testing system (LANHE CT 2001 A) with a voltage range from 2.5 to 4.0 V (1 C = 170 mA g^−1^) at 60 °C. All batteries were assembled in an Ar-filled glove box with oxygen and H_2_O content less than 0.1 ppm.

## 3. Results and Discussion

### 3.1. Preparation of PLiSSPSI-g-SiO_2_ Nanoparticles

Lithium (4-styrenesulfonyl)(phenylsulfonyl) imide (LiSSPSI) as single-ion monomer used in this work was prepared according to the previous report [19], and the ^1^H NMR spectrum and peak assignments of LiSSPSI are presented in Appendix A. RAFT agent, 2-methyl-2-[(dodecylsulfanylthiocarbonyl)sulfanyl]propanoic acid-2-(2-pyridyldithio) ethyl ester (py-ss-DTPA), was synthesized via an esterification reaction between 2-methyl-2-[(dodecylsulfanylthiocarbonyl)sulfanyl]propanoic acid (DTPA) and 2-(2-pyridinyldithio)ethanol (PDE) [20], and the ^1^H NMR spectra and peak assignments of DTPA, PDE and py-ss-DTPA are shown in Appendix A, respectively. Py-ss-PLiSSPSI was synthesized through RAFT polymerization using py-ss-DTPA as RAFT agent and 2,2′-azobis(2-methylpropionitrile) (AIBN) as initiator (Figure 1). The ^1^H NMR spectrum and peak assignments of py-ss-PLiSSPSI are shown in Appendix A. Based on the ^1^H NMR results, the average number of repeating units is calculated, and size exclusion chromatography (SEC) curves and the molecular weight dispersity of the polymer are shown in Appendix A.

Thiol-functionalized silica particles (SiO_2_-SH) were prepared following the previous work [23], and py-ss-PLiSSPSI brushes on silica particles (PLiSSPSI-*g*-SiO_2_) were prepared after thiol-disulfide exchange reaction between SiO_2_-SH and py-ss-PLiSSPSI (Figure 1). Pyridine-2-thione yielded in the exchange reaction has a characteristic absorption at 375 nm in DMF in UV-vis spectra (Figure 1a). Based on a standard curve (Figure 1b), the amount of pyridine-2-thione is obtained and the grafting density of py-ss-PLiSSPSI on silica particles is calculated to be 0.67 mg·g^−1^ [20].

FT-IR spectra of SiO_2_-SH, py-ss-PLiSSPSI and PLiSSPSI-*g*-SiO_2_ are presented in Figure 2, and the characteristic peak of -SH in SiO_2_-SH measured by Raman spectroscopy is shown in Appendix A. As observed in FT-IR spectra, the peak at 3061 cm^−1^ belongs to the stretching vibration of aromatic heterocycle of py-ss-PLiSSPSI. The peaks at 2930 and 746 cm^−1^ can be assigned to the saturated C-H bonds and S-N bonds of PLiSSPSI segments, respectively. Peaks at 1667, 1443 and 1407 cm^−1^ are originated from the vibration of benzene rings of PLiSSPSI. The strong absorption peak appears at 470 cm^−1^ corresponds to the stretching vibration of -Si-O-Si- of SiO_2_. Compared to pristine SiO_2_-SH, PLiSSPSI-*g*-SiO_2_ owns the characteristic absorption peaks of py-ss-PLiSSPSI, which indicates that the pyridyl disulfide functionalized polymer py-ss-PLiSSPSI was grafted onto SiO_2_ successfully.

The surface structures of SiO_2_-SH and PLiSSPSI-*g*-SiO_2_ are further investigated by XPS (Figure 3). Figure 3a shows the XPS deconvoluted C 1s peaks. For the SiO_2_-SH, the peak at 284.80 eV is originated from the C-C bonds. The deconvoluted peaks at 284.80, 286.12 and 288.82 eV exist in the PLiSSPSI-*g*-SiO_2_ correspond to the C-C, C-S and -O-C=O bonds, respectively [24]. The deconvoluted O 1s spectra are shown in Figure 3b. For the SiO_2_-SH, one peak presents at 530.60 eV, which matches the Si-O bonds [25]. In comparison, the PLiSSPSI-*g*-SiO_2_ shows two deconvoluted peaks at 530.60 and 531.70 eV, which can be assigned to the Si-O bonds and the SO_2_-N bonds, respectively [25,26]. In Figure 3c, the S 2p peaks can be fitted into two spin orbit doublets with a splitting of 1.18 eV and a 2:1 area ratio. The S 2p spectra of the SiO_2_-SH and PLiSSPSI-*g*-SiO_2_ both show deconvoluted peaks at 163.50 (2p_3/2_) and 164.48 eV (2p_1/2_), which are originated from the S-H or S-C bonds. For the PLiSSPSI-*g*-SiO_2_, and a small peak at around 167.33 eV corresponds to the structure of -SO_2_-C- [26]. Figure 3d shows the deconvoluted spectra of the Si 2p region. Both SiO_2_-SH and PLiSSPSI-*g*-SiO_2_ show only one peak at 100.80 eV originated from the Si-O bonds [25]. The existence of C-C, C-S, -O-C=O and -SO_2_-C- bonds demonstrates the py-ss-PLiSSPSI are successfully grafted onto the surface of SiO_2_ nanoparticles.

Figure 4a shows the SEM of SiO_2_-SH and PLiSSPSI-*g*-SiO_2_. The diameters of SiO_2_-SH and PLiSSPSI-*g*-SiO_2_ are around 600 nm and 700 nm, respectively, consistent with the DLS results in methanol (Appendix A). The surface for the SiO_2_-SH appears smooth, but the surface of PLiSSPSI-*g*-SiO_2_ particles is rougher. This indicates that the py-ss-PLiSSPSI is successfully grafted onto the SiO_2_ surface. The chemical compositions and element distributions of SiO_2_-SH and PLiSSPSI-*g*-SiO_2_ particles are characterized by scanning transmission electron microscopy (STEM). The results demonstrate that both Si and O are homogeneously distributed (Figure 4b). The EDS analysis indicates that the content of the S is less in SiO_2_-SH particles than that of PLiSSPSI-*g*-SiO_2_ (Figure 4c). The N (1%) is observed in PLiSSPSI-*g*-SiO_2_ particles. Therefore, XPS, SEM and EDS results indicate that PLiSSPSI are successfully grafted onto SiO_2_.

### 3.2. Composite Solid Polymer Electrolytes Based on Poly(ethylene oxide) and PLiSSPSI-g-SiO_2_ Nanoparticles

Nowadays, the application of PEO solid electrolytes is still limited by their poor mechanical properties and low ionic conductivity at room temperature [8]. Efforts have been made to decrease the crystallinity of PEO in order to improve the ionic conductivity of PEO-based solid electrolytes, such as fabricating PEO with cross-linking structure [27], blending PEO with other polymers [28] and forming PEO-based block copolymers [29,30]. Inorganic nanoparticles, e.g., SiO_2_ and TiO_2_ have been introduced into PEO-based composite solid electrolytes (CSE) to enhance their mechanic properties [31,32]. In this work, a kind of single-ion conduction polymer modified SiO_2_ (PLiSSPSI-*g*-SiO_2_) is utilized as the functional filler to prepare composite solid electrolytes PLi-*g*-SiCSE-x, where x represents the weight content of PLiSSPSI-*g*-SiO_2_ nanoparticles. The concentration of PLiSSPSI-*g*-SiO_2_ nanoparticles in the composites is varied between 2 and 10 wt% with respect to the total weight.

Figure 5 showed the XRD patterns and DSC curves of PEO-based CSE with various concentrations of PLiSSPSI-*g*-SiO_2_ nanoparticles, as well as the PEO/LiTFSI ([EO]: [Li^+^] = 20) for comparison. In Figure 5a, for the PEO/LiTFSI, two sharp diffraction peaks appear at 2θ = 19 and 23^o^, respectively, indicative of the presence of highly crystalline phase in PEO [33]. Upon addition of the PLiSSPSI-*g*-SiO_2_ nanoparticles into PEO/LiTFSI, the intensities of these two XRD peaks decrease, and the peaks become broad compared to PEO/LiTFSI. This can be attributed to the random distribution of PLiSSPSI-*g*-SiO_2_ nanoparticles into PEO interrupts the ordered arrangement of PEO polymer chains, and consequently increases the content of amorphous phase of CSE. In Figure 5b, for PEO/LiTFSI electrolyte, the *T_g_* was −49.3 °C and the *T_m_* is 27.9 °C. With the rise in the content of PLiSSPSI-*g*-SiO_2_ nanoparticles, the *T_g_* of PLi-*g*-SiCSE drops from -50.8 °C (2 wt%) to −54.3 °C (6 wt%), then increases to −50.0 °C (10 wt%). Meanwhile, the *T_m_* of PLi-*g*-SiCSE decreases from 25.6 °C (2 wt%) to 22.8 °C (6 wt%), then increases to 23.6 °C (10 wt%). Both *T_g_* and *T_m_* of PLi-*g*-SiCSEs are lower than those of PEO/LiTFSI electrolyte, which indicates that the introduction of PLiSSPSI-*g*-SiO_2_ nanoparticles can increase the content of amorphous phase of PLi-*g*-SiCSEs, consist with the XRD results. The PLi-*g*-SiCSE-6 exhibits the lowest *T_g_* and *T_m_*, which suggests that its content of the amorphous phase is the highest. The high content of amorphous phase is beneficial to promoting the movement of EO segment and to enhancing the ionic conductivity of the CSE.

Figure 6 shows the SEM images of the surface and cross-section morphologies of the prepared CSEs containing PEO/LiTFSI and different amounts of PLiSSPSI-*g*-SiO_2_ nanoparticles. The surface of all CSEs is smooth, while the cross-section of the CSEs containing PLiSSPSI-*g*-SiO_2_ nanoparticles is smoother than that of PEO/LiTFSI CSE. In addition, PLi-*g*-SiCSEs-10 exhibits nanoparticle aggregations in film as shown in Figure 6(d3) compared to PLi-*g*-SiCSEs-2 (Figure 6(b3)) and PLi-*g*-SiCSEs-6 (Figure 6(c3)).

The mechanical properties of PLi-*g*-SiCSEs were also investigated. The stress-strain behavior of PLi-*g*-SiCSEs was studied (Figure 7), and the tensile strength and elongation of the PLi-*g*-SiCSEs was characterized. The results are summarized in Table 1.

The PEO/LiTFSI electrolyte shows a tensile strength of 0.97 MPa and maximum elongation of 830%. In the presence of PLiSSPSI-*g*-SiO_2_ nanoparticles, the tensile strength of membranes increases to 3.63 MPa (6 wt%), then decreases to 3.37 MPa (10 wt%). With the increased content of the PLiSSPSI-*g*-SiO_2_ nanoparticles, the breaking elongation decreases to 657% (2 wt%), then increased to 1147% (10 wt%). This phenomenon is due to the plasticized effect and conforms to the stress and strain law of solid membrane. Especially, the PLi-*g*-SiCSEs-6 membrane shows the most favorable mechanical properties, with a tensile strength of 3.63 MPa and a breaking elongation of 911%.

The ionic conductivity of PLi-*g*-SiCSEs at different temperature is calculated based on the intercept of the high-frequency region measuring by electrochemical impedance spectroscopy (EIS) (Appendix A and Appendix A), and the conductivity of CSEs at 60 °C is summarized in Appendix A. Figure 8 compares the conductivities as a function of temperature for the CSEs with various contents of PLiSSPSI-*g*-SiO_2_ nanoparticles, as well as the PEO/LiTFSI CPE. All the PLi-*g*-SiCSEs exhibit higher ionic conductivity than that of PEO/LiTFSI CPE in all temperature. This might be caused by the chain entanglement between LiSTFSI on PLiSSPSI-*g*-SiO_2_ and PEO that reduces the crystallinity of PEO, and improves the compatibility of heterogeneous interface. A lower *T_g_* is favorable to increase the amorphous phase and thus enhance the ionic conductivity of the CSE due to the relaxation of polymer chains. Hence, based on the *T_g_* values shown in Figure 5b, the higher ionic conductivity is obtained in PLi-*g*-SiCSE-6. At 60 °C, with the increase of the content of PLiSSPSI-*g*-SiO_2_ nanoparticles, the ionic conductivity increased from 0.17 mS cm^−1^ of PLi-*g*-SiCSE-2 to 0.22 mS cm^−1^ of PLi-*g*-SiCSE-6, then decreased to 0.13 mS cm^−1^ of PLi-*g*-SiCSE-10. With low amounts of PLiSSPSI-*g*-SiO_2_ nanoparticles, the single-ion conducting polymer-modified SiO_2_ can disperse well in PEO/LiTFSI CSE, which can create more amorphous regions in CSEs, therefore the ionic conductivity is increased. However, higher amounts of PLiSSPSI-*g*-SiO_2_ nanoparticles are inclined to agglomerate within the PEO, which could cause the obstruction of ion transport. The ionic conductivity of PLi-*g*-SiCSE-6 reaches 0.22 mS cm^−1^ at 60 °C (Appendix A), higher than those reported CSEs containing PEGMA-modified SiO_2_ [34] and PEGMA/single-ion monomer-modified SiO_2_ and Al_2_O_3_ [15] owing to the better dispersion caused by PLiSSPSI, while lower than the CSEs containing polymer-grafted SiO_2_ [35,36] and neat polymeric nanoparticles[17,18] which can be attributed to the much larger particle size (around 700 nm as shown in Figure 4 and Appendix A).

The lithium-ion transference number (t_Li+_) is also a key parameter of SPE. The t_Li+_ for the PLi-*g*-SiCSEs and the classic PEO/LiTFSI ([EO]: [Li^+^] = 20) CSE at 60 °C are measured by a combination of impedance and potentiostatic polarization method. The impedance spectra and the *I-t* curve of PLi-*g*-SiCSE-6 are measured using the Li/PLi-*g*-SiCSE-x/Li symmetric cell at 60 °C (Figure 9a). The results of t_Li+_ for CSEs are calculated by Equation (2) and are presented in Figure 9b. The t_Li+_ of PEO/LiTFSI, PLi-g-SiCSE-2, PLi-g-SiCSE-6, and PLi-g-SiCSE-10 are 0.32, 0.48, 0.77 and 0.51, respectively. Obviously, all the t_Li+_ of PLi-g-SiCSEs are higher than that of PEO/LiTFSI CSE, which is caused by the grafting of PLiSSPSI on the surface of SiO_2_ that can not only inhibit the movement of anions in CSE, but also promote the dissociation of lithium salt and increase the mobility of Li^+^. The t_Li+_ of PLi-*g*-SiCSE varies with the content of PLiSSPSI-*g*-SiO_2_ nanoparticles, which indicates that well-dispersed PLiSSPSI-*g*-SiO_2_ nanoparticles can improve the t_Li+_ effectively, and that the agglomeration resulted by high concentration of nanoparticles would reduce the t_Li+_.

Furthermore, the electrochemical window is crucial for developing high voltage and high-power lithium batteries. Traditional PEO/LiTFSI CSE is usually oxidized at a relatively low voltage, which limits the energy density [37]. The electrochemical window is measured by linear sweep voltammetry using SS|PLi-g-SiCSE-x|Li cell at 0.1 mV s^−1^ and the results are shown in Figure 10. All the PLi-*g*-SiCSEs are stable up to 4.2 V vs. Li/Li^+^. The electrochemical window for PLi-*g*-SiCSE-6 even reaches 5.6 V, indicating that it possesses good electrochemical stability and feasibility in high-voltage lithium batteries.

The PLi-*g*-SiCSEs are sandwiched between two lithium metal electrodes to assemble all-solid-state Li/Li symmetrical batteries. Figure 11a shows the time-dependent plating/stripping profile of the cell with PLi-*g*-SiCSE-6 as electrolyte over 480 h at a constant current density of 0.2 mA·cm^−2^ and at 60 °C. A stable overpotential of 27 mV demonstrate that reversible lithium plating/stripping performance can be achieved in Li|PLi-*g*-SiCSE-6|Li cell, and no short circuit is observed. Figure 11b exhibits an enlarged plot of the lithium plating/peeling cycling curves of Li|PLi-*g*-SiCSE-6|Li symmetric cell at 70–80 h. The overpotential voltage is around 27 mV, indicating its superior interfacial stability with the lithium electrodes.

Figure 12 shows the rate capability and initial charge-discharge curves of LFP|PLi-*g*-SiCSE-6|Li at 60 °C. The reversible discharge specific capacities are 140, 139, 132 and 121 mAh·g^−1^ at current densities of 0.1, 0.2, 0.5 and 1 C, respectively (Figure 12a). After five cycles, the specific capacity at 0.1 C recovered to 140 mAh·g^−1^, which indicates good rate performance of the PLi-*g*-SiCSE-6 in lithium batteries. Typical charge-discharge potential curves at different rates show that the voltage gaps between the charge and discharge plateaus are 0.09 V, 0.15 V, 0.30 V, and 0.53 V at 0.1 C, 0.2 C, 0.5 C, and 1 C, respectively (Figure 12b) [38]. The approximate linear variation of the voltage gap with current density further illustrates the selective Li^+^ conductivity of PLi-g-SiCSE-6, which improves the rate capability of the assembled cell. This phenomenon is attributed to the presence of organic-inorganic hybrid nanoparticles that enhance the interfacial compatibility between the membrane and the electrode, thereby reducing polarization [36].

Meanwhile, the high-rate cycling performance of the LFP|PLi-*g*-SiCSE-6|Li battery was investigated (Figure 13a). The results show that the battery exhibits a specific capacity of 124 mAh·g^−1^ at 1 C (60 °C) and with a coulombic efficiency of 100%. After 130 cycles, the specific capacity decayed to 80%, while the coulombic efficiency remains 100%. As shown in Figure 13b, the polarization potential is within 0.4 V after 20 cycles. After 130 cycles, the polarization potential increases to 0.5 V which is possibly caused by the exfoliation of the cathode active material and the loss of the interfacial contact after long cycles at high current densities [16].

## 4. Conclusions

In this work, a single-ion conducting polymer modified SiO_2_ (PLiSSPSI-*g*-SiO_2_) was served as a functional filler in PEO SPE. The SiO_2_-SH nanoparticles were prepared via sol-gel process, then PLiSSPSI-*g*-SiO_2_ nanoparticles were synthesized through thiol-disulfide exchange reaction of the SiO_2_-SH and py-ss-PLiSSPSI. The uniform distribution of single-ion conducting polymer layer on the surface of SiO_2_ reduces the crystallinity of the PEO effectively which can promote the transference of Li^+^. The composite solid electrolyte with 6 wt% of PLiSSPSI-*g*-SiO_2_ nanoparticle exhibits a relatively high ionic conductivity (0.22 mS·cm^−1^) at 60 °C, a wide electrochemical window (5.6 V vs. Li/Li^+^) and a large t_Li+_ (0.77). PLi-*g*-SiCSE-6 also shows superior mechanical properties with the tensile strength of 3.63 MPa and breaking elongation of 911%. Meanwhile, when charged and discharged at 0.2 mA·cm^−2^ current densities, the Li|PLi-*g*-SiCSE-6|Li cell could cycle for more than 480 h without short circuit. The improved properties of the PLi-*g*-SiCSE-6 endow the LFP|PLi-*g*-SiCSE-6|Li battery with good cycling performance (the specific discharge capacity reaches 140 mAh·g^−1^ at 0.1 C) and rate capability (average specific discharge capacities of 140, 139, 132 and 121 mAh·g^−1^ at the C-rates of 0.1, 0.2, 0.5 and 1 C, respectively). Furthermore, the all-solid-state LFP|PLi-*g*-SiCSE-6|Li battery demonstrates stable cycling performance for 120 cycles at 1 C and 60 °C. Hence, the single-ion conducting polymer functionalized nanoparticles possess great potential as novel fillers to promote the utilization of corresponding CSEs used in the next generation of all solid-state high-voltage lithium-ion batteries. Moreover, the size effect of single-ion conducting polymer modified nanoparticles on the performances of CSEs is expected to be further investigated.

## Data Availability

Data openly available in a public repository.

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
