# Peer review of "Enhanced Electrochemical Performance of PEO-Based Composite Polymer Electrolyte with Single-Ion Conducting Polymer Grafted SiO2 Nanoparticles"

_polymers, 2023, doi:10.3390/polym15020394_

Round 1
Reviewer 1 Report
The work is scientifically sound, and while there is new information here, the methodology novelty is somewhat adequate.
In a present state the manuscript submitted is adequate and to be recommended for publication with minor correction.
Listed of the corrections and question:
1. Reduce the number of references for [7-11] or explain in text. Don’t only mention many references without explanation.
2. Section “Materials”, for same producer, must list the chemical together. i.e : methanol, ethanol,…. from Macklin.
3. All chemicals and instruments must be writing in standardised. If mentioned the country of origin, so that all must be mentioned.
4. Figure 1(b), must have “error bar”.
Author Response
Thanks for your kindly comments, according to your suggestion, I have made the modifications as follows:
- I have shortened the references [7-11] to [7-9] to illustrate the utilization of functional nanoparticles in SPEs.
- I have already put the materials supplying from the same supplier together.
- The countries of the instruments are all added.
- For Figure 1(b), we have added the error bar in the stand curve according to your advice.
Reviewer 2 Report
This paper deals with the development of f single-ion conducting polymer modified nanoparticles for solid state electrolyte for use in Li batteries.
Certainly there is some merit in this paper but the paper is terse to read since many results are discussed and also scattered in supporting information and the main paper. The striking feature of this paper is the claim that the operational potential window is enlarged which may have some appeal to researchers.
The synthesis, characterization and electrochemical studies all are needed but the reader may find difficult to comprehend all of them together.
The paper deserves to be documented
Author Response
Thanks for your kindly comments.
Reviewer 3 Report
In this work, a pyridyl disulfide-terminated polymer was synthesized via RATF and was grafted on to silica, which was further loaded in PEO at different loading to prepare solid electrolyte for Li-ion battery. 6% loaded composite was found to exhibit the optimum result.
The work is nice and insightful. However, the following points should be addressed before publication:
1. Equivalent circuit and the values should be included for EIS.
2. Signature of thiol in FTIR peak assignment should be mentioned.
Author Response
Thanks for your kindly comments, according to your suggestion, I have made the modifications as follows:
- The equivalent circuit and the values of SS|PLi-g-SiCSE-6|SS are added and shown in Figure S9 and Table S1.
- For the characteristic peak of thiol in SiO2-SH, it cannot be observed by FT-IR spectroscopy, we used Raman spectroscopy to identify it, the result is shown in Figure S7.
Reviewer 4 Report
This manuscript report on the synthesis, characterization and electrochemical response of new composite electrolyte materials for lithium-ion batteries. This is a topic of utmost fundamental and applied importance in the field of materials for energy. The research strategy proposed here, i.e., grafting a single-ion conducting polymer on SiO2 nanoparticles to be embedded in a PEO solid electrolyte, is original and interesting and the results appear to be promising. Based on that, the manuscript deserves to be published. However, the work contains a number of major weakness that must be corrected before considering publication:
- The interpretation of the C1s spectrum of PLiSSPSI-g-SiO2 seems strange (p. 7). In particular, assigning the 286.1 eV contribution to carboxylic groups is in contrast to the whole literature for the position of such groups (around 289 eV; see for instance the XPS spectra of PMMA or PET). In the present interpretation, the carbon atom in the carboxylic group is considered to be more electron-rich than carbon atoms bound to sulfur, which is in clear contradiction with the electronegativity of carbon, sulfur, and oxygen. This point must be addressed.
- The interpretation of the S2p spectra is awfully wrong. The two components in the spectrum of SiO2-SH are not due to the presence of two different chemical species. Instead they correspond to the spin-orbit splitting (2p1/2 and 2p3/2 components) that exists for all p, d, and f core lines in XPS. This is a huge mistake, which shows that the authors do not have even the most basic knowledge of XPS.
- The comparison of the diameter of the particles before and after grafting implies that the thickness of the polymer layer is 50 nm. Considering the extreme situation in which the polymer chains would be fully extended perpendicular to the particle surface, this means that the chains are at least 50 nm long. Is this consistent with the molecular weight of the polymer ? This point must be addressed.
- The authors ascribe the improvement of the conductivity of the composites to a change in Tg (p. 12). However, that change is only of a few degrees and those Tgs are all around -50°C, which is very far below the temperature range of the conductivity measurements (from +25 to +80°C). So it is highly doubtful that the increase in conductivity is related to the Tg. Most probably, it is related to the decrease in crystallinity, which clearly appears in Fig. 5a. This point must be addressed.
Technical point:
Is it relevant to give values for elongation at break with 0.1% ‘precision’ (Table 1) ?
To conclude, this manuscript should be considered for publication in Polymers only when these important issues have been thoroughly addressed.
Author Response
Thanks for your kindly comments, according to your suggestion, I have made the modifications which is shown in the attachment.

Round 2
Reviewer 4 Report
The authors have properly addressed the comments and suggestions. the manuscript can now be published.